# Simulated Microgravity Created Using a Random Positioning Machine Induces Changes in the Physiology of the *Fusarium solani* Species Complex

**DOI:** 10.3390/microorganisms10112270

**Published:** 2022-11-16

**Authors:** Maurine D’Agostino, Anne-Lyse Babin, Marie Zaffino, Jean-Pol Frippiat, Marie Machouart, Anne Debourgogne

**Affiliations:** 1UR 7300 SIMPA, Stress Immunity Pathogens Laboratory, Faculty of Medicine, Lorraine University, 9 Avenue de la Forêt de Haye, F-54500 Vandœuvre-lès-Nancy, France; 2Service de Microbiologie, CHRU de Nancy, Hôpitaux de Brabois, 11 Allée du Morvan, 54511 Vandœuvre-lès-Nancy, France

**Keywords:** *Fusarium solani* species complex (FSSC), simulated microgravity, ground-based facilities, random positioning machine, biofilm, growth, germination

## Abstract

*Fusarium* is a phytopathogenic fungus involved in human pathology and is present in space stations. It is essential to understand the effects of microgravity on the physiology of this fungus to determine the potential risks to the health of crew members and to propose the necessary countermeasures. This study aimed to determine changes in the physiological parameters of the *Fusarium solani* species complex under simulated microgravity generated using a random positioning machine (RPM) and phenotypic approaches. We observed increased growth, spore production, and germination while biofilm production was reduced under RPM exposure. These in vitro data show the importance of further studying this fungus as it has been repeatedly demonstrated that microgravity weakens the immune system of astronauts.

## 1. Introduction

The space environment modifies numerous human physiological functions. A recent study of medical data collected from 46 astronauts who spent 6 months on the International Space Station (ISS) showed that 46% encountered immunological problems such as hypersensitivities and infections [1]. The ISS harbors a variety of microorganisms, originating from Earth, experimental components, and the microbiota of crew members [2]. Among them, the presence of the fungus *Fusarium* has been described in surface samples and condensation liquids of the Mir (Russian space station) and ISS stations [3,4]. Recently, *Fusarium* strains have been isolated on the surface of the ISS dining table but also in lesions observed on a plant cultivated at this station [5,6]. The presence of this pathogen in space could potentially cause health problems for the crew because it is now clear that spaceflights weaken the immune system [7,8,9]. This fungus could also induce degradation of embarked equipment [10]. It is therefore essential to determine physiological changes in *Fusarium* when it grows under simulated microgravity. On Earth, the *Fusarium* genus is a transkingdom pathogen. It is a phytopathogenic fungus but also an emerging fungus involved in the origin of invasive fungal infections in immunocompromised human subjects, characterized by fungemia or skin lesions [11]. The *Fusarium solani* species complex (FSSC) represents 60% of human fusariosis in humans [12].

Some studies have analyzed the adaptation of fungi to microgravity (Table 1), but they mainly investigate commensal yeasts such as *Candida albicans* [13]. Phenotypic approaches or global transcriptional profiles have been presented in the literature; however, no adaptation mechanism has been proposed for fungi of medical interest. Different models have been used to study microgravity, including experiments during space flights and simulated microgravity models. Among these, we have chosen the random positioning machine (RPM), which is a valid alternative for conducting examinations on the influence of the force of gravity in a fast and straightforward manner [14].

Due to its presence in space stations and its major pathogenicity in humans, we studied the effect of simulated microgravity on the physiological aspects of the *Fusarium solani* species complex. Indeed, it is essential to understand these effects to determine the potential risks to the health of crew members and to propose the necessary countermeasures. To our knowledge, no data are currently available for this pathogen. In a first descriptive approach, essential given the current lack of knowledge, we evaluated the physiological parameters of growth, germination, and ability to generate biofilms under RPM exposure.

## 2. Materials and Methods

### 2.1. Isolates

Eight *FSSC* isolates (CBS 115660, CBS 124630, CBS 124896, CBS 124895, CBS 124889, CBS 224.34, CBS 102824, and CBS 117608) were used for this study (Table 2) and cultured on Sabouraud medium at 30 °C before the experiments described in the following paragraphs. These strains present different genotypes and minimum inhibitory concentrations (MICs) determined by both the Clinical and Laboratory Standards Institute (CLSI) and E-test methods [25]. 

### 2.2. Microgravity Simulation

The RPM (Figure 1) of the Gravitational Experimental Platform for Animal Models (GEPAM) [26,27], a European Space Agency (ESA) ground-based facility, was used to simulate microgravity (8). The continuous change in the orientation of objects with respect to the gravity vector can generate effects comparable to those of real microgravity (7). The principle of the RPM is that over time the trajectory of the gravity vector points in all directions is calculated by an appropriate algorithm, meaning that after a longer time, the mean gravity reaches theoretical zero. To use this principle, the following RPM settings were used throughout the incubation phase [28,29,30]: random speed, direction, and interval, with an angular velocity of rotation between 48 and 72 degrees per second. For each experiment, test samples were at the center of the rotation metallic platform. Indeed, simulated microgravity is obtained for samples placed approximately 10 cm around the center of rotation [31]. Control samples were placed on the base of the RPM, either without rotation to benefit from the same environmental conditions or with 1 g agitation to allow a homogeneous fungal culture.

### 2.3. Macroscopic Characteristics of Cultures

Two types of agar media were used to study the macroscopic aspect of cultures under RPM exposure: oatmeal agar (OA) (Millipore, Merck, Germany), recommended to describe *Fusarium* macroscopy [32] and Roswell Park Memorial Institute medium (RPMI) (BioMérieux, Marcy-l’Étoile, France). On each solid medium in Petri dishes (55 mm), three inoculation points of the same strain were created with 10 μL of a suspension at 10^6^ conidia/mL. All strains were studied. The media, exposed to RPM or static 1 g control, were incubated at 35 °C for 72 to 96 h in the dark. The macroscopic appearance of fungal colonies was observed with the naked eye and with a binocular magnifier. Different aspects such as the texture, thickness, and color of the colonies were compared between static 1 g control and simulated microgravity conditions.

### 2.4. Growth and Spore Production Evaluation

The growth was evaluated by optical density measurements at 600 nm for 24 h in microplates. Each well (RPM and static 1 g) was incubated with 5 × 10^4^ conidia/mL and incubated at 30 °C in the dark. 

To evaluate the spore production between the two gravity conditions, 5 mL tubes of liquid Sabouraud medium (Millipore, Merck, Germany) were inoculated with 10^6^ conidia/mL. After 24 and 48 h of incubation at 30 °C, for each condition (RPM and static 1 g), the number of colonyforming units (CFUs) was counted on Sabouraud agar (Millipore, Merck, Germany). 

The experiment was performed twice for all 8 strains.

### 2.5. Germination

For the germination assay, conidia were collected by adding 5 mL of sterile water containing 0.1% Tween 80 to the Petri dish after peeling conidia off three times with a rake. The recovered suspension was centrifuged for 5 min at 2000× *g* and conidia were counted on Kova slides (Kova International, Garden Grove, CA, USA), and diluted in liquid Sabouraud medium to 10^6^ conidia/mL [33]. Each 5 mL tube was incubated at 30 °C in the dark either in the RPM or with shaking for the 1 g control. Germinated spores were counted by microscopy using Kova slides at 24 and 48 h intervals and the percentage of germination was determined. The experiment was performed twice for all 8 strains.

### 2.6. Biofilm

#### 2.6.1. Biofilm Formation

Petaka culture chambers were used to carry out fungal cultures in liquid medium without air bubbles (Petaka basic kit, Celartia, Columbus, OH, USA). It is a rectangular chamber (85.5 × 127.5 × 5.0 mm) designed for cell culture that contains an automatic gas diffusion device. For biofilm formation, 4 mL of standardized cell suspension at 10^6^ conidia/mL produced in Sabouraud medium was injected into the device followed by 25 mL of Sabouraud culture medium. The chamber was then incubated for 90 min at 37 °C flat and without shaking for cell adhesion. Then, the cultures were incubated for 72 h at 37 °C in the dark under either RPM exposure or 1 g control conditions. The media was removed, and pieces of the chambers (1 cm^2^) were cut for quantitative and qualitative analysis of the biofilm. These analyses were performed for the CBS 124630 strains in duplicate and the experiment was performed five times.

#### 2.6.2. Biofilm Quantification

A semiquantitative approach to biofilm synthesis was carried out using the XTT ((2,3-bis-(2-méthoxy-4-nitro-5-sulfophényl)-2H-tétrazolium-5-carboxanilide) method described by Chandra [34]. Petaka pieces were placed in a twelve-well cell culture plate with 4 mL of phosphate-buffered saline (PBS). Then, 100 μL of XTT at 1 mg/mL was added to each well, and 8 μL of 1 mM menadione was added. The plate was covered with aluminum foil and incubated overnight at 37 °C with shaking. The medium was recovered, transferred to a 15 mL propylene tube, and centrifuged for 5 min at 4 °C and 3500× *g*. The supernatant was collected and its absorbance was measured at 490 nm.

#### 2.6.3. Qualitative Biofilm Analysis

A qualitative approach to biofilm synthesis was carried out by confocal microscopy according to Chandra’s protocol [34]. It involves two markers, FUN-1 (2-chloro-4-(2,3-dihydro-3-methyl-(benzo-1,3-thiazol-2-yl)-methylidene)-1-phenylquinolinium iodide) and concanavalin-A. FUN-1 is a dye used to monitor cell viability in fungi and yeasts. Petaka pieces were placed in a twelve-well cell culture plate. A mixture containing 2 mL of PBS, 10 μL of Con-A 5 mM, and 2 μL of FUN-1 10 mM was added to each well for 30 min at 37 °C. Then, the pieces were removed from the wells and mounted on a slide. The excitation and emission wavelengths were 543 and 560 nm for FUN-1 and 488 and 505 nm for CON-A, respectively. Biofilm observation was performed using a Leica TCS SP5 X AOBS confocal microscope (Wetzlar, Germany) mounted on a DMI6000 and piloted by LAS AF of the Imaging and Cellular Biophysics Platform of the University of Lorraine using a 40-immersion objective in water. The thickness of the biofilm was determined using the “Z-stack” function. Confocal microscopy images were analyzed using Image J software 1.53 t developed by the National Institute of Health (USA).

### 2.7. Statistical Analysis

Data analysis was performed using XLSTAT 1194 (XLSTAT statistical and data analysis solution, New York, NY, USA. https://www.xlstat.com, 1 March 2021). Differences between groups were determined using Student’s *t* tests and considered statistically significant at *p* values ≤ 0.05.

## 3. Results

### 3.1. Macroscopic Aspects of Cultures

No significant effect due to RPM exposure was observed for any of the experiments.

### 3.2. Growth and Spore Production Evaluation

A growth difference was observed at 24 h of culture between the 1 g control (OD of 0.563) and RPM exposure (OD of 0.670) conditions (*p* = 0.001).

Under simulated microgravity, spore production was twice as high as in the 1 g control condition (Figure 2). Indeed, after inoculation with the same quantity of cells, at 24 h of incubation, 73 and 31 CFU were observed under simulated microgravity and 1 g control conditions, respectively (*p* < 0.001). At 48 h of incubation, the number of colonies was 111 and 46, respectively, under the conditions described above (*p* < 0.001).

### 3.3. Germination

The germination ratio was higher under simulated microgravity than under 1 g control conditions (Figure 3). At 24 h of incubation, the percentage of germinated spores was 89% in 1 g control samples and 95% in RPM-exposed samples (*p* < 0.01); at 48 h, the values were 84 and 91%, respectively (*p* < 0.01). Under RPM exposure, multiple germination tubes on several areas of the spore were observed, while they were absent in the control condition (Figure 4).

### 3.4. Biofilm Formation

The quantity of synthesized biofilm, evaluated by the XTT method, seemed lower under simulated microgravity by comparison to 1 g controls in each experiment (Figure 5), but did not reach statistical significance. A 30% decrease in metabolic activity was observed between the controls (mean OD of 0.300) and the strains subjected to simulated microgravity (mean OD of 0.210), but the *p* value was 0.156.

Confocal microscopy observation showed that the biofilm formed in 1 g controls was denser than the ones formed under RPM exposure for each experiment (Figure 6). The thickness determined using the “Z-stack” function (Figure 5) showed that biofilms were thinner under simulated microgravity conditions. The average thickness of the biofilm in the control condition was between 37 μm and 70 μm, while it was between 30 μm and 40 μm under RPM exposure. The average thickness for the control condition (50 μm) was greater than that under RPM exposure (34 μm); a difference of 32% was observed (*p* = 0.027). 

## 4. Discussion

This study aimed to understand the impact of microgravity, simulated by an RPM, on the physiological parameters of *Fusarium*, a filamentous fungus that has already shown pathogenicity in plants in space conditions [5,6]. Furthermore, *Fusarium* has been associated with material degradation [10].

Experimentation in real microgravity is expensive and scarcely available; thus, a variety of platforms with ground-based facilities provide experimental conditions on Earth comparable to real microgravity [31]. In this study, microgravity is simulated using an RPM. The potential bias of this model related to sample positioning was excluded by comparing the results obtained from samples differing in position on the rotation plate (Appendix A).

For the mushroom *Pleurotus ostreatus*, macroscopic modifications of fruiting body development are obvious under simulated microgravity. Growth against the direction of gravity is observed when the fungus is placed under static conditions (fixed to the ground), whereas the mushroom fruits radially under simulated microgravity [35]. In our study, slight differences were observed in the macroscopic appearance of the colonies in culture, without revealing a consensual modification. The same observation was made for other filamentous fungi (*Aspergillus niger*, *Cladosporium herbarum*, *Ulocladium chartarum*, *Basipetospora halophila*), whose growth took place onboard the ISS [36]. For yeasts, the appearance of the colonies was not modified in real microgravity conditions [18], whereas under simulated microgravity conditions, the appearance of hyper irregular wrinkle morphology was described [19].

In our study, we showed an increase in fungal growth and spore production under RPM exposure. Comparable data were described for *Candida albicans*, *Saccharomyces cerevisiae* and *Aspergillus fumigatus* under spaceflight or simulated microgravity conditions [17,18,20,23,24].

We also observed an increase in spore germination under RPM exposure with spores presenting multiple germination tubes. The same observation was made for *Aspergillus niger* and *Penicillium,* whereas no differences were observed for *Aspergillus fumigatus* [24,37]. For *Candida albicans*, most studies showed an increase in yeast filamentation [15,16,17,19]. Budding patterns were also modified in yeasts (*Candida albicans*, *Saccharomyces cerevisiae*) with a higher frequency of multiple buds and sometimes the formation of yeast clusters [13,15,22,23].

By two complementary approaches, we demonstrated a decrease in *Fusarium* biofilm formation under RPM exposure while *Candida albicans* grown under simulated microgravity presented a more complex biofilm [19]. However, it was recently shown that this microorganism forms a biofilm with fewer retained cells in the biofilm when cultivated on the ISS [18].

Thus, our study showed that under simulated microgravity conditions, the filamentous fungus *Fusarium solani* presents stronger growth and greater germination, which raises the hypothesis of a greater fungal load in pathology. On the other hand, it appears that this fungus has less need to resort to its biofilm form in this environment, suggesting better accessibility for immune cells or antifungal molecules. To test these hypotheses, studies of virulence and susceptibility in an in vivo model are needed. This type of study is essential in the field of space research to establish the necessary countermeasures to preserve the health of the crew as well as the integrity of the installations. In addition, in the medical field, these models allow us to understand the potential modifications of microorganisms in low shear areas of the body such as microvilli of epithelial cells in the gastrointestinal, urogenital, and respiratory tracts [2,38].

To conclude, we have shown that simulated microgravity prompts physiological changes in the filamentous fungus *Fusarium solani*.

## Figures and Tables

**Figure 1 microorganisms-10-02270-f001:**
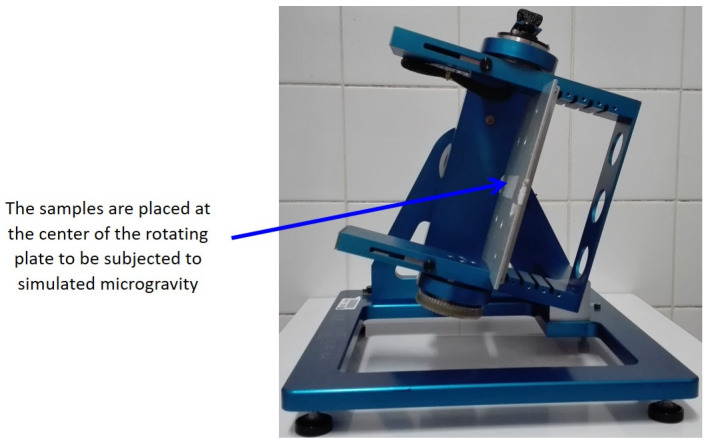
Photograph of the random positioning machine (personal collection). This equipment allows the exposure of biological samples to simulated microgravity. This machine rotates the sample randomly around the Earth’s gravity vector resulting in an average net force close to zero, thus simulating microgravity.

**Figure 2 microorganisms-10-02270-f002:**
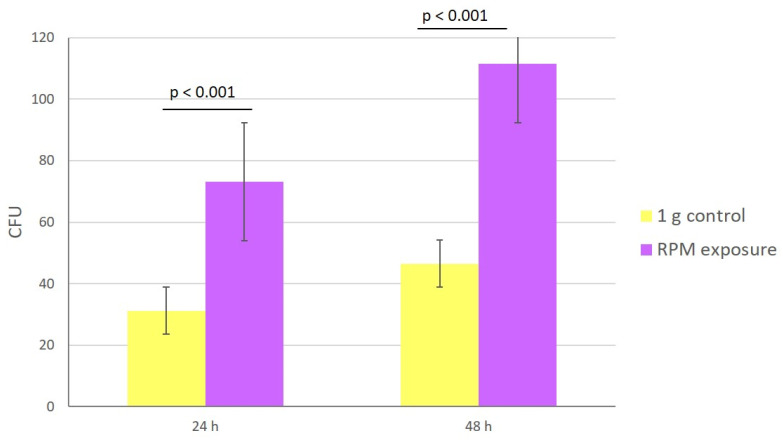
Impact of microgravity on fungal growth. Number of CFUs after 24 and 48 h of culture under simulated microgravity (purple) or 1 g control (yellow) conditions. The experiment was performed twice for all 8 strains subjected to 1 g control or RPM conditions. Data are the mean ± SEM. Statistically significant differences were observed using the Student’s *t* test.

**Figure 3 microorganisms-10-02270-f003:**
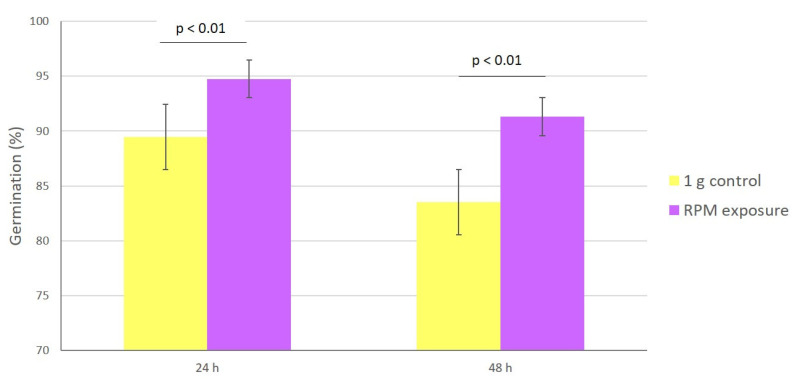
Impact of microgravity on spore germination. Percentage of germination under simulated microgravity (purple) and 1 g control (yellow) conditions at 24 and 48 h of incubation. The experiment was performed twice for all 8 strains subjected to 1 g control or RPM conditions. Data are the mean ± SEM. Statistically significant differences were observed using the Student’s *t* test.

**Figure 4 microorganisms-10-02270-f004:**
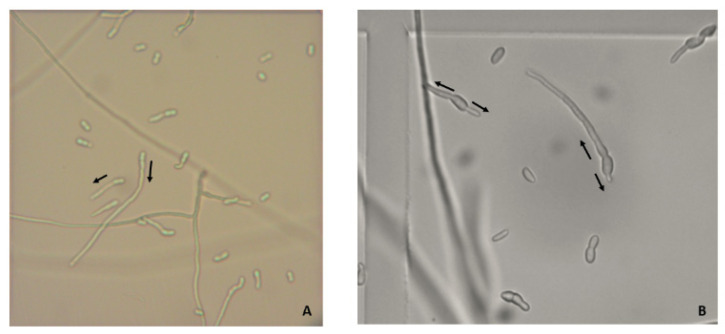
Multiple germination tubes on spores in microgravity. Multiple germination tubes observed using optic microscopy on spores of the CBS 124889 strain obtained 1 g control (**A**) or under simulated microgravity (**B**) at 48 h (arrow to visualize the direction of germination) (×200).

**Figure 5 microorganisms-10-02270-f005:**
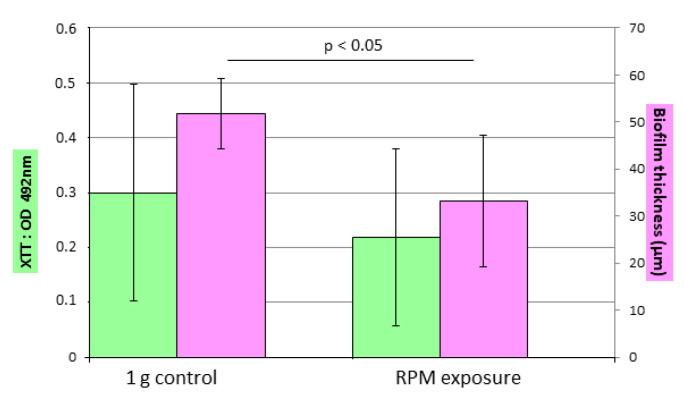
Impact of microgravity on biofilm formation. Evaluation of biofilm formation using the XTT test (green) and confocal microscopy to measure biofilm thickness (pink) in 1 g control and simulated microgravity conditions. Duplicates of the CBS 124630 strain were subjected to 1 g control or RPM conditions. Experiments were performed five times. Data are the mean ± SEM. Statistically significant differences were observed using the Student’s *t* test.

**Figure 6 microorganisms-10-02270-f006:**
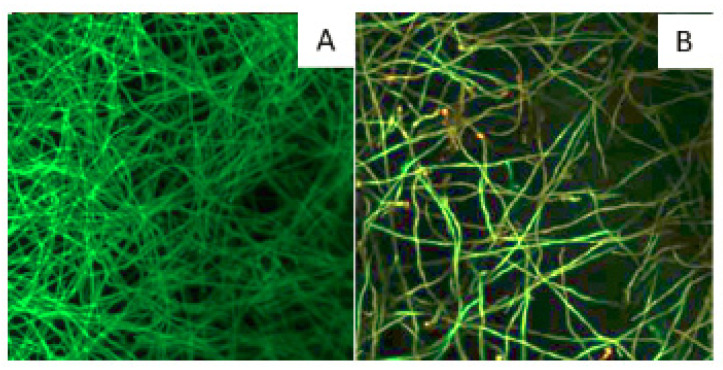
Impact of microgravity on biofilm density. Observation, using confocal microscopy, of the biofilm formed under 1 g control (**A**) and simulated microgravity (**B**) conditions after labeling with FUN-1 (intracellular labeling in red) and ConA (labeling of the cell wall in green) (×400). Example of biofilm formation by the CBS 124630 strain after 72 h of culture in the RPM or at 1 g.

**Table 1 microorganisms-10-02270-t001:** Review of the effects of simulated or real microgravity on fungi of medical interest.

Fungus	Model	Observed Effect	Reference
*Candida albicans*	HARV	Increase in filamentation and modification of budding.	[15]
*Candida albicans*	Spaceflight	Modification of budding and clusters formation.	[13]
*Candida albicans*	Spaceflight 2D Clinostat	Decline in virulence. Dimorphic shift with branching and formation of filaments.	[16]
*Candida albicans*	Rotary cell culture system	Increase in growth and filamentation. Decreased viability in the presence of antifungals.	[17]
*Candida albicans*	Spaceflight	Increase in growth and amphotericin resistance.	[18]
*Candida albicans*	HARV	Increase in filamentation. Generation of biofilm with a more complex structure. Modification of the morphology of the colonies. Increased viability of yeasts exposed to amphotericin B.	[19]
*Saccharomyces cerevisiae*	Diamagnetic levitation	Increase in growth.	[20]
*Saccharomyces cerevisiae*	Clinostat	Reduction in longevity.	[21]
*Saccharomyces cerevisiae*	Rotary cell culture system	Modification of budding.	[22]
*Saccharomyces cerevisiae*	HARV	Modification of budding and clusters formation.	[23]
*Aspergillus fumigatus*	ISS	Increase in growth and virulence.	[24]

ISS: International Space Station. HARV: high aspect ratio vessel.

**Table 2 microorganisms-10-02270-t002:** Genotype and origin of the strains of the *Fusarium solani* species complex used.

Collection Number	Genotype	Location	Geography
CBS 115660	21 d	Potatoes	Egypt
CBS 124630	3+4 bbbb	Skin	France
CBS 124896	11 i	Blood	Belgium
CBS 124895	5 o	Skin	France
CBS 124889	2 d	Nail	France
CBS 224.34	1 b	Toe nail	Cuba
CBS 102824	25 d	Plated litter fragment	Columbia
CBS 117608	6 f	Arm lesion dermis	Turkey

The genotype of FSSC is noted with number and letters.

## Data Availability

Data available on request due to restrictions. The data presented in this study are available on request from the corresponding author. The data are not publicly available due to privacy.

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
