# Peer review of "Simulated Microgravity Created Using a Random Positioning Machine Induces Changes in the Physiology of the *Fusarium solani* Species Complex"

_microorganisms, 2022, doi:10.3390/microorganisms10112270_

Round 1
Reviewer 1 Report
Dear Authors
The manuscript was written in rather poor English and it has some serious issues:
1- The abstract is too short and it lacks a description of the methodology and the main outcome of the work.
2- The introduction has to be reconstructed in a logical way and you have to show the gap they want to fill and your motivation.
3- A lot of abbreviations were introduced more than one time while other abbreviations were not introduced at all.
4- There are some repetitions in the discussion and some hypotheses lack scientific evidence.
5- There are no clear conclusions to wrap up and show the main outcome of the work.
Please find my comments in the attached pdf file
Good luck

Author Response
Please see the attachmen

Reviewer 2 Report
Please provide the abstract with more significant data
Reviewer 3 Report
The manuscript is concerning a research work entitled “Simulated microgravity created using a random positioning machine induces changes in the physiology of Fusarium solani species’ complex”.
Q1: Abstract is very concise so you must add 3 to 4 lines in order to make the study bit more interesting for readers.
Q2: Lines No: 33 and 34 give reported information for human pathology and list a few disorders that have been caused and reported by the Fusarium genus in humans.
Q3: Line No: 44 to 45 no adaptation mechanism has been proposed for fungi of medical interest. Could you come to assume a clue for adoptable mechanisms?
Q4: In heading 2.1 give a brief methodology of isolates, how strains were cultured, and which media was used to screen out isolates?
As you just mentioned locality and source from where You have collected
You make your methodology strong by citing mentioned article for culturing isolates… https://www.mdpi.com/2309-608X/8/7/753
Q5: In Figures 2 and 5 especially why standard error bars showed huge differences what condition has created a huge difference?
The overall manuscript is well-written, and the data are well interpreted
TRANSLATE with x English
Arabic | Hebrew | Polish |
Bulgarian | Hindi | Portuguese |
Catalan | Hmong Daw | Romanian |
Chinese Simplified | Hungarian | Russian |
Chinese Traditional | Indonesian | Slovak |
Czech | Italian | Slovenian |
Danish | Japanese | Spanish |
Dutch | Klingon | Swedish |
English | Korean | Thai |
Estonian | Latvian | Turkish |
Finnish | Lithuanian | Ukrainian |
French | Malay | Urdu |
German | Maltese | Vietnamese |
Greek | Norwegian | Welsh |
Haitian Creole | Persian |
Round 2
Reviewer 3 Report
Thanks to all the authors for incorporating all the suggested changes. I believe the manuscript has been sufficiently improved for publication.
TRANSLATE with x EnglishArabic | Hebrew | Polish |
Bulgarian | Hindi | Portuguese |
Catalan | Hmong Daw | Romanian |
Chinese Simplified | Hungarian | Russian |
Chinese Traditional | Indonesian | Slovak |
Czech | Italian | Slovenian |
Danish | Japanese | Spanish |
Dutch | Klingon | Swedish |
English | Korean | Thai |
Estonian | Latvian | Turkish |
Finnish | Lithuanian | Ukrainian |
French | Malay | Urdu |
German | Maltese | Vietnamese |
Greek | Norwegian | Welsh |
Haitian Creole | Persian |